

# Serum PCSK9 is a novel serological biomarker for the diagnosis and prognosis of pancreatic cancer

Ying Xu*, Yongfeng Xu*, Yang Yang, Zhiwei Zhang, Qunli Xiong and Qing Zhu

Division of Abdominal Tumor Multimodality Treatment, Cancer Center, West China Hospital, Sichuan University, Chengdu, Sichuan, China
* These authors contributed equally to this work.

## ABSTRACT

**Background:** Although CA19-9 is an essential blood biomarker of pancreatic cancer (PC), its sensitivity and specificity are limited for early detection.

**Methods:** We analyzed the serum proprotein convertase subtilisin/kexin type 9 (sPCSK9) in PC patients, benign disease groups (BDG), and healthy controls (HC) by ELISA.

**Results:** Consistently, sPCSK9 was considerably lower in PC patients than in HC ($Z = -2.546$, $P < 0.05$), and sPCSK9 in PC patients was statistically significantly higher than in BDG ($Z = -5.457$, $P < 0.001$). sPCSK9 was linked to the invasion of lymph nodes ($\chi^2 = 6.846$, $P < 0.01$). According to ROC curves, combining sPCSK9 with CA19-9 could potentially enhance the diagnostic capability of CA19-9 in early-stage PC patients. Furthermore, the low sPCSK9 group ($n = 41$) exhibited statistically significantly prolonged overall survival compared to the high sPCSK9 group ($n = 15$), with median survival times of 27 months (95% CI [17.59–36.41]) and 11 months (95% CI [7.21–14.79]), respectively ($P = 0.022$).

**Conclusion:** The diagnostic performance of CA19-9 for early-stage PC patients could be improved by combining sPCSK9 with CA19-9. Moreover, the higher sPCSK9 group has a significantly shorter overall survival rate.

## INTRODUCTION

Pancreatic cancer (PC) is one of the most malignant gastrointestinal tumors, with a dismal prognosis and high mortality. Recent trends indicate a notable surge in the incidence of PC, indicating its emergence as a significant contributor to cancer-related mortalities on a global scale (*Huang et al., 2021*; *Klein, 2021*). Unfortunately, merely 15–20% of patients can undergo surgery, primarily due to the non-specific and vague nature of early symptoms in the majority of cases (*Tempero et al., 2019*). Hence, there is an urgent need to discover a new biomarker for early PC diagnosis to improve efficiency and accuracy. The simplicity and expeditious nature of serum biomarker detection render it the preferred method for pancreatic cancer screening (*Stoffel, Brand & Goggins, 2023*). While many

Corresponding author
Qing Zhu,
newzhuqing1972@scu.edu.cn

biomarkers are related to pancreatic cancer, identifying a single index has certain limitations.

Reprogramming energy metabolism has been accepted as an emerging hallmark of cancer (*Hanahan & Weinberg, 2011*), and lipid metabolism is a major metabolic alteration in cancer (*Cheng et al., 2018*). Reprogramming of lipid metabolism is involved in cellular proliferation, invasion, and metastasis by activating specific enzymes and regulating various oncogenic signal pathways (*Butler et al., 2020*; *Snaebjornsson, Janaki-Raman & Schulze, 2020*). In addition, Cancer cells exhibit increased lipid and cholesterol avidity, leading to abnormal lipid accumulation in the tumor microenvironment, which affects the cancer immune response (*Corn, Windham & Rafat, 2020*; *Yu et al., 2021*). Proprotein convertase subtilisin/kexin type 9 (PCSK9), a pivotal enzyme in lipid metabolism, exerts its influence on cholesterol metabolism through the facilitation of low-density lipoprotein receptor (LDLR) degradation (*Seidah & Prat, 2022*). Emerging data have revealed that PCSK9 plays a key role in lipid-centric cardiovascular diseases and has novel roles in cancer cell proliferation, apoptosis, invasion, and metastasis (*Bhattacharya et al., 2021*). More recently, research has shown that PCSK9 is involved in both anti-tumor immunity and immunological response. Tumor cell-derived PCSK9 downregulates LDLR levels in CD8+ T cells (CTLs) and attenuates TCR signaling, thereby inhibiting the antitumor activity of CTLs (*Yuan et al., 2021*). Furthermore, PCSK9 interacts with MHC I, leading to its degradation and reducing membrane surface MHC I levels, facilitating evasion of CTL recognition (*Liu et al., 2020*). Overexpression of PCSK9 has been observed in several cancer types indicating that PCSK9 holds promise as a diagnostic and prognostic biomarker, such as colorectal cancer (*Wang et al., 2022*; *Wong et al., 2022*), hepatocellular carcinoma (*Jin et al., 2023*), and gastric cancer (*Xu et al., 2020*). PCSK9 is a potential factor in tumor cell proliferation, invasion, and metastasis (*Bhattacharya et al., 2021*).

Although the role and expression characteristics of PCSK9 in various tumor cells have been well established, its specific expression profile in pancreatic cancer and its correlation with clinical prognosis remain inadequately elucidated. Hence, we performed an observational cohort trial to identify the clinical utility of PCSK9 in pancreatic cancer in this context.

# MATERIALS & METHODS

## Study design and ethical considerations

The study was conducted in March 2020 and July 2023 in West China Hospital of Sichuan University (Chengdu, China). The Pathology Department of West China Hospital made the diagnosis and graded the pancreatic cancer, and the American Joint Committee on Cancer (AJCC) criteria standard defined TNM staging. The enrollment criteria of the pancreatic cancer group were: (1) pathologically confirmed pancreatic cancer; (2) no history of other primary malignancies; (3) have not received any treatment. The exclusion criteria were: (1) incomplete medical records; (2) history of other cancer-related therapies; and (3) current inflammatory diseases. The control population was comprised of benign disease patients and normal individuals. The benign disease patients were clinically

diagnosed with benign disease of the pancreas, including chronic pancreatitis, benign tumor of the pancreas, *etc*. Normal individuals were conducted with no signs of diseases.

The Ethics Committee at Sichuan University's West China Hospital gave its approval for the study, which was carried out in compliance with the Declaration of Helsinki (1189, 20220325). Before being recruited, all participants were apprised of the ethical information relevant to this study and written informed consent was obtained. Patients diagnosed with pancreatic cancer were followed up *via* telephone interviews or outpatient clinic visits, allowing for the acquisition of overall survival (OS) information from all participants. All PC patients were followed up until October 2023.

## Samples collection

The study involved the enrollment of 289 blood samples: the PC group consisted of 118 patients, while the control group consisted of 171 patients, which included 97 benign disease groups (BDG, including chronic pancreatitis (CP), $n = 62$; pancreatic neuroendocrine tumor, $n = 10$; intraductal papillary mucinous neoplasm (IPMN), $n = 9$; cystadenoma, $n = 7$; solid pseudopapillary tumor of pancreas, $n = 6$; others, $n = 3$) and 74 healthy controls (HC). All blood samples were collected before surgical resection, chemotherapy, or radiotherapy. Any hemolyzed blood samples were excluded. A total of 5 ml of fasting blood obtained from each participant was collected into tubes containing heparin sodium and then stored at 4 °C for transfer to a laboratory. After centrifuging serum samples for 10 min at 3,000 × rpm at 4 °C, they were split into 2 to 4 aliquots and stored at −80°C until analysis. Additional clinical data were gathered from the West China Hospital case management system, including age, sex, TNM stage, grade, serum lipid level, and serum level of CA19-9.

## Measurement of sPCSK9 levels

sPCSK9 levels were quantified using enzyme-linked immunosorbent assay (ELISA) kits (Cat#EH384RB, Thermo Fisher Scientific, Waltham, MA, USA) following the manufacturer's instructions. Absorbance values were measured at a wavelength of 450 nm using a 96-well microplate reader. Subsequently, the sPCSK9 concentration was determined by comparing the obtained result to the standard sample and constructing a standard curve.

## Immunohistochemistry

The study involved the enrollment of 48 tissue samples, which were obtained from the pathology platform of West China Hospital and have been reviewed by the ethics department. After paraffin embedding, the thickness of the tissue samples was sectioned into 3 mm sections deparaffinized in xylene and rehydrated through different concentrations of alcohols (100%, 95%, 90%, and 80%). Endogenous peroxidase activity was blocked with hydrogen peroxide (3%) at 37 °C for 15 min. The slides were subjected to antigen retrieval by pressure cooking in citrate buffer (pH 6.0) for 16 min. A polyclonal anti-PCSK9 antibody (Cat. No 27882-1-AP, Proteintech, Chicago, IL, USA) was used as the reaction at a dilution of 1:20 at 37 °C for 1h. After washing the sections in PBS, we

incubated them with biotinylated secondary antibody for 40 min at 37 °C. Diaminobenzidine (DAB) solution as the chromogen and hematoxylin solution as the counterstain. For the negative control, 1% BSA/PBS was used in place of the primary antibody and was processed in the same manner.

The slides were captured digitally, stored as high-resolution images files using the Slideview VS200 slide scanner (Olympus, Tokyo, Japan) and analyzed using the OlyVIA visualizer version 2.4 (Olympus, Tokyo, Japan). The IHC score of these areas: (% area of weak staining) + (2 × % area of moderate staining) + (3 × % area of strong staining). Then, we received a score between 0 and 300.

## Statistical analysis

All variables were subjected to descriptive statistics, and the Shapiro–Wilk test assessed the data's normality. Based on the results, the statistical methods were selected as follows. The $\chi^2$ test was used to classify data. For normally distributed count data, the independent samples T-test was employed. For abnormally distributed count data, the non-parametric Mann-Whitney U test was applied. The study compared the expression levels of sPCSK9 and CA19-9 in PC patients and the control group using non-parametric Mann–Whitney U tests. The study utilized the receiver operating characteristic (ROC) curve to assess the sensitivity and specificity of sPCSK9 and CA19-9. The respective areas under the curves (AUCs) with 95% confidence intervals (CI) were also evaluated to determine the cutoff line that provides the optimal diagnostic accuracy and likelihood ratios. We also compare the performance of different ROC curves by using the DeLong test. The study assessed the correlation between serum levels of sPCSK9 and CA19-9 using Spearman rank correlation, and the correlation between serum levels of sPCSK9 and HDL using Pearson rank correlation. Survival was contrasted using log-rank tests by creating survival curves through the Kaplan-Meier method. We used the SPSS version 23.0 (IBM Corp, Chicago, IL, USA) to analyze all statistics. A statistically significant difference was defined as a $P$-value < 0.05. Finally, we used the GraphPad Prism V.9.5 software (GraphPad Software, La Jolla, CA, USA) to draw figures.

# RESULTS

## Participants characteristics

The baseline clinical characteristics of the patients and benign disease groups are presented in Table 1. Between the two groups, demographic traits were well-balanced. Among the 118 PC patients, stratification based on AJCC stages resulted in four groups: stage I ($n = 33$), stage II ($n = 22$), stage III ($n = 33$), and stage IV ($n = 30$).

## Serum level of PCSK9 in PC patients and control group

Figure 1 depicts the sPCSK9 in both the PC and control groups. The PCSK9 serum concentration was considerably lower in the PC group compared to the healthy control (HC) group (Z = −2.546, $P < 0.05$). Conversely, the sPCSK9 levels in the PC group were significantly higher than in the benign disease group (BDG) (Z = −5.457, $P < 0.001$). Furthermore, sPCSK9 in the BDG group was significantly lower compared to the HC

**Table 1 Clinicopathological characteristics of recruited participants.**

| Variable | Pancreatic cancer (n = 118) | Control | | P |
|---|---|---|---|---|
| | | Benign pancreatic tumor (n = 35) | Chronic pancreatitis (n = 62) | |
| **Age, year** | | | | |
| <65 | 73(61.9%) | 32(91.4%) | 56(90.3%) | P < 0.05 |
| ≥65 | 45(38.1%) | 3(8.6%) | 6(9.7%) | |
| **Sex, n(%)** | | | | |
| Male | 85(72.0%) | 15(42.9%) | 45(72.6%) | P = 0.113 |
| Female | 33(28.0%) | 20(57.1%) | 17(27.4%) | |
| **BMI** | | | | |
| Mean ± SD | 21.885 ± 2.935 | 22.752 ± 3.693 | 21.528 ± 2.874 | P = 0.842 |
| **Smoke, n(%)** | | | | |
| Yes | 48(40.7%) | 8(22.9%) | 23(37.1%) | P = 0.187 |
| No | 70(59.3%) | 27(77.1%) | 39(62.9%) | |
| **Drink, n(%)** | | | | |
| Yes | 32(27.1%) | 5(14.3%) | 18(29.0%) | P = 0.569 |
| No | 86(72.9%) | 30(85.7%) | 44(71.0%) | |
| **Diabetes, n(%)** | | | | |
| Yes | 18(15.3%) | 3(8.6%) | 12(19.4%) | P = 0.966 |
| No | 100(84.7%) | 32(91.4%) | 50(80.6%) | |
| **Hypertension, n(%)** | | | | |
| Yes | 30(25.4%) | 10(28.6%) | 14(22.6%) | P = 0.909 |
| No | 88(74.6%) | 25(71.4%) | 48(77.4%) | |
| **Triglycerides** | | | | |
| Median (IQR) | 1.33(1.010–1.775) | 1.38(0.960–1.620) | 1.295(0.928–1.910) | P = 0.940 |
| **Cholesterol** | | | | |
| Median (IQR) | 4.26(3.425–4.870) | 4.1(3.580–4.610) | 3.84(3.263–4.545) | P = 0.156 |
| **HDL** | | | | |
| Mean ± SD | 1.033 ± 0.371 | 1.193 ± 0.297 | 0.986 ± 0.368 | P = 0.542 |
| **LDL** | | | | |
| Mean ± SD | 2.403 ± 0.828 | 2.418 ± 0.638 | 2.0998 ± 0.790 | P = 0.072 |
| **T stage, n(%)** | | | | |
| T1-2 | 50(42.4%) | | | |
| T3-4 | 68(57.6%) | | | |
| **Lymph node invasion** | | | | |
| Positive | 42(35.6%) | | | |
| Negative | 76(64.4%) | | | |
| **Metastasis** | | | | |
| Positive | 30(25.4%) | | | |
| Negative | 88(74.6%) | | | |
| **TNM stage(AJCC)** | | | | |
| I | 33(28.0%) | | | |

(Continued)

| Table 1 (continued) | | | | |
|---|---|---|---|---|
| | Pancreatic cancer | Control | | P |
| Variable | (n = 118) | Benign pancreatic tumor (n = 35) | Chronic pancreatitis (n = 62) | |
| II | 22(18.6%) | | | |
| III | 33(28.0%) | | | |
| IV | 30(25.4%) | | | |
| Tumor differentiation | | | | |
| Well | 3(2.5%) | | | |
| Moderate | 45(38.1%) | | | |
| Poor | 39(33.1%) | | | |
| Missing | 31(26.3%) | | | |

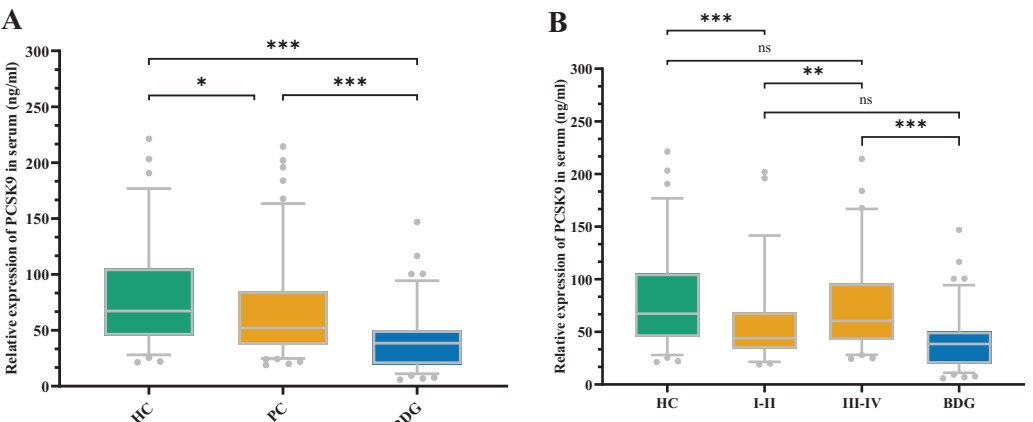

**Figure 1 The expression of sPCSK9 levels.** (A) sPCSK9 levels in the healthy control group (HC), pancreatic cancer group (PC), and benign disease group (BDG). HC (n = 74), PC (n = 118), BDG (n = 97). (B) sPCSK9 levels in the HC, BDG, and different staging groups. HC (n = 74), BDG (n = 97), I/II (n = 55), III/IV (n = 63). *P < 0.05, **P < 0.01 and ***P < 0.001. ns, none significance.

group (Z = −6.853, P < 0.001) (Fig. 1A). Patients with early PC had considerably lower sPCSK9 levels than those with advanced PC (Z = −2.751, P < 0.01) (Fig. 1B). In addition, the sPCSK9 levels of patients diagnosed with early PC were significantly lower than those of the HC group (Z = −3.674, P < 0.001), and sPCSK9 levels of patients with advanced PC were much higher than those in the BDG group (Z = −5.845, P < 0.001) (Fig. 1B).

## Correlations between clinicopathological characteristics and sPCSK9 levels in PC patients

We performed the nonparametric Mann–Whitney U test to compare sPCSK9 levels among clinical subgroups. Our analysis indicated that the group without lymph node invasion had significantly lower sPCSK9 levels compared to the positive groups (Z = −2.799, P < 0.01). However, the sPCSK9 levels had no significant differences among
the remaining clinical factors, such as age, sex, diabetes, sex, hypertension, tumor invasion, or metastasis (Fig. S1).

Further, we separated PC patients into two groups based on the sPCSK9 cut-off value (54.83 ng/ml), with high and low sPCSK9 levels, and then used the $\chi^2$ test to interpret the correlations between sPCSK9 levels and Clinicopathological features (Table 2). We found that the level of sPCSK9 had a strong correlation with drinking ($P = 0.008$), lymph node invasion ($P = 0.009$), and TNM stage ($P = 0.022$). However, the sPCSK9 level did not significantly correlate with other clinical features, such as age, sex, smoking, diabetes, hypertension, tumor invasion, or metastasis ($P > 0.05$).

## Diagnostic value of sPCSK9 and CA19-9

We assessed the diagnostic value of sPCSK9 in detecting PC (Table S1), particularly early-stage PC (stage I-II) by receiver operating characteristic (ROC) curve analysis. ROC analysis determined the cutoff values, sensitivity, and specificity of CA19-9 and sPCSK9 levels. When comparing patients with early-stage PC to the HC group, the best cut-off levels for CA19-9 and sPCSK9 were 18.6 U/mL and 54.4 ng/mL, respectively. The area under the curve (AUC) values for CA19-9 *versus* sPCSK9 were 0.725 (95% CI [0.604–0.846], sensitivity 71.7% and specificity 82.4%) and 0.689 (95% CI [0.598–0.781], sensitivity 65.5% and specificity 67.6%), as illustrated in Fig. 2A. When comparing patients with early-stage PC to the BDG, the best cut-off levels for CA19-9 and sPCSK9 were 42.70 U/mL and 27.4 ng/mL, respectively. The AUC values for CA19-9 *versus* sPCSK9 were 0.706 (95% CI [0.595–0.818], sensitivity 52.2% and specificity 92.6%) and 0.651 (95% CI [0.563–0.738], sensitivity 92.7% and specificity 35.1%), as illustrated in Fig. 2B. Then, we evaluated the diagnostic performance of the combination of CA19-9 and sPCSK9 in early-stage PC patients and controls. When comparing the AUC of the combination between early-stage PC and HC, the AUC value for the combined group was statistically significantly higher than the CA19-9 group (0.875, 95% CI [0.802–0.949], sensitivity 78.3% and specificity 84.3%; Fig. 2A) ($Z = 3.176$, $P < 0.05$). When comparing the AUC of the combination between early-stage PC and BDG, the AUC value for the combined group also was higher than that of the CA19-9 group (0.762, 95% CI [0.668–0.855], sensitivity 65.2% and specificity 81.9%; Fig. 2B). But it's not statistically significant ($Z = 1.697$, $P = 0.11$). The sensitivity and specificity of the sPCSK9 and CA19-9 in detecting early-stage PC are shown in Table 3.

In addition, we explored the relationship between sPCSK9 and CA19-9. The test kit has limitations, and any result higher than or equal to 1,000 U/ml will be disregarded for CA19-9 measurement. After that, we used Spearman's correlation analysis to determine the relationships between sPCSK9 and CA19-9 levels in each patient. The results showed a negative level trend ($r = 0.257$, $P < 0.05$, Fig. 3).

## Prognostic value of sPCSK9 levels before treatment for PC patients

To evaluate the prognostic significance of sPCSK9 in PC patients, we identified a sPCSK9 cutoff of 54.83 ng/ml, which approximates the optimal threshold observed when comparing PC patients with HC. Utilizing this cutoff, we stratified PC patients into two

**Table 2 Characteristics of patients with pancreatic cancer.**

| Characteristics | Number | PCSK9 | | $\chi^2$ | P |
|---|---|---|---|---|---|
| | | Low | High | | |
| **Age, year** | | | | | |
| <65 | 73(61.9%) | 36(49.3%) | 37(50.7%) | 1.869 | P = 0.172 |
| ≥65 | 45(38.1%) | 28(62.2%) | 17(37.8%) | | |
| **Sex, n(%)** | | | | | |
| Male | 85(72.0%) | 47(55.3%) | 38(44.7%) | 0.137 | P = 0.712 |
| Female | 33(28.0%) | 17(51.5%) | 16(48.5%) | | |
| **Smoke, n(%)** | | | | | |
| Yes | 48(40.7%) | 23(47.9%) | 25(52.1%) | 1.302 | P = 0.254 |
| No | 70(59.3%) | 41(58.6%) | 29(41.4%) | | |
| **Drink, n(%)** | | | | | |
| Yes | 32(27.1%) | 11(34.4%) | 21(65.6%) | 6.979 | P < 0.01 |
| No | 86(72.9%) | 53(61.6%) | 33(38.4%) | | |
| **Diabetes, n(%)** | | | | | |
| Yes | 18(15.3%) | 11(61.1%) | 7(38.9%) | 0.404 | P = 0.525 |
| No | 100(84.7%) | 53(53.0%) | 47(47.0%) | | |
| **Hypertension, n(%)** | | | | | |
| Yes | 30(25.4%) | 19(63.3%) | 11(36.7%) | 1.341 | P = 0.247 |
| No | 88(74.6%) | 45(51.1%) | 43(48.9%) | | |
| **T stage, n(%)** | | | | | |
| T1-2 | 50(42.4%) | 30(60.0%) | 20(40.0%) | 1.161 | P = 0.281 |
| T3-4 | 68(57.6%) | 34(60.0%) | 34(60.0%) | | |
| **Lymph node invasion** | | | | | |
| Positive | 42(35.6%) | 16(38.1%) | 26(61.9%) | 6.846 | P < 0.01 |
| Negative | 76(64.4%) | 48(63.2%) | 28(36.8%) | | |
| **Metastasis** | | | | | |
| Positive | 30(25.4%) | 13(43.4%) | 17(56.7%) | 1.927 | P = 0.165 |
| Negative | 88(74.6%) | 51(58.0%) | 37(42.0%) | | |
| **TNM stage(AJCC)** | | | | | |
| I-II | 55(46.6%) | 36(65.5%) | 19(34.5%) | 5.222 | P < 0.05 |
| III-IV | 63(53.4%) | 28(44.4%) | 35(55.6%) | | |
| **Tumor differentiation** | | | | | |
| Well-Moderate | 48(40.7%) | 31(64.6%) | 17(35.4%) | 0.604 | P = 0.437 |
| Poor | 39(33.0%) | 22(56.4%) | 17(43.6%) | | |
| Missing | 31(26.3%) | | | | |

cohorts: high sPCSK9 and low sPCSK9. Interestingly, the low sPCSK9 group (n = 41) exhibited a significantly prolonged overall survival compared to the high sPCSK9 group (n = 15) (27 months (95% CI [17.59–36.41]) vs. 11 months (95% CI [7.21–14.79]), P = 0.022), as depicted in Fig. 4.

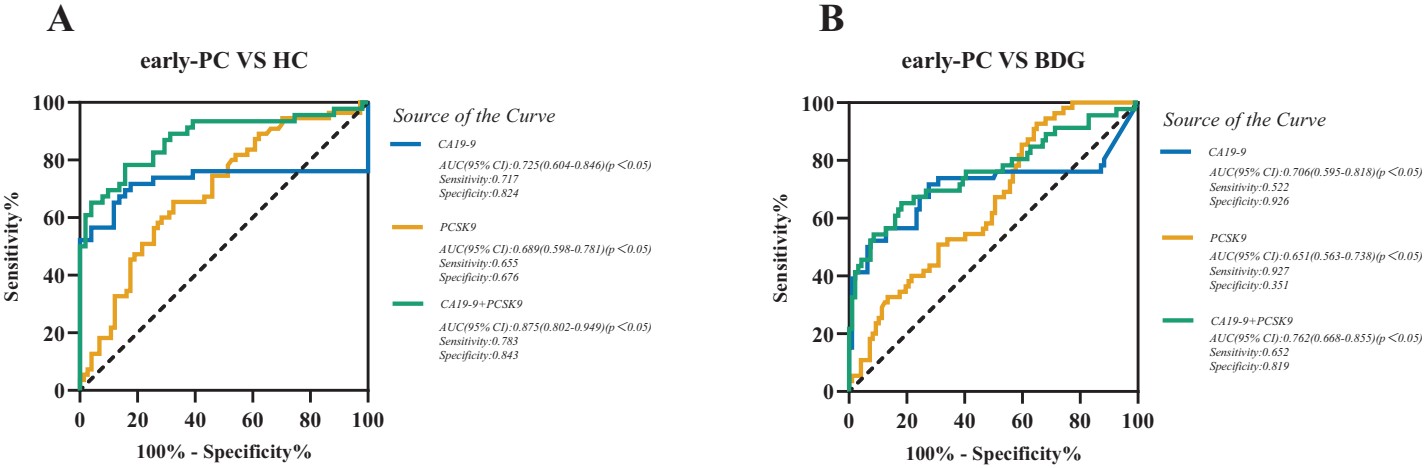

**Figure 2 Receiver operating characteristic curve (ROC) analyses of sPCSK9 (orange line), CA19-9 (blue line), and CA19-9 and PCSK9 combined group (green line) in the diagnosis of early-stage PC.** (A) ROC curve for CA19-9, PCSK9, and combined group for early-stage PC patients *versus* HC; (B) ROC curve for CA19-9, PCSK9, and combined group for early-stage PC patients *versus* BDG.

**Table 3 A diagnostic performance of sPCSK9 and CA19-9 in detecting patients with early-stage PC.**

| Biomarkers | AUC | 95%CI | Cut-off value | Sensitivity(%) | Specificity(%) | P |
|---|---|---|---|---|---|---|
| **Early-PC *vs* HC** | | | | | | |
| CA19-9 | 0.725 | [0.604–0.846] | 18.6 U/mL | 71.7 | 82.4 | <0.05 |
| PCSK9 | 0.689 | [0.598–0.781] | 54.4 ng/mL | 65.5 | 67.6 | <0.05 |
| PCSK9+CA19-9 | 0.875 | [0.802–0.949] | – | 78.3 | 84.3 | <0.05 |
| CA19-9 *vs* PCSK9+CA19-9 | | | | | | <0.05 |
| **Early-PC *vs* BDG** | | | | | | |
| CA19-9 | 0.706 | [0.595–0.818] | 42.7 U/mL | 52.2 | 92.6 | <0.05 |
| PCSK9 | 0.651 | [0.563–0.738] | 27.4 ng/mL | 92.7 | 35.1 | <0.05 |
| PCSK9+CA19-9 | 0.762 | [0.668–0.855] | – | 65.2 | 81.9 | <0.05 |
| CA19-9 *vs* PCSK9+CA19-9 | | | | | | 0.11 |

**Note:**
Abbreviations: AUC, areas under the curve.

## Immunohistochemistry staining of PCSK9 in human pancreatic cancer tissues

To assess PCSK9 protein expression in pancreatic cancer tissues, immunohistochemistry (IHC) was carried out in pancreatic tissues containing normal pancreatic tissues and tumor tissues. Statistical analysis indicated a significant increase of PCSK9 expression in tumor tissues (t = 7.693, $P < 0.01$) compared to normal tissues (Fig. 5).

## DISCUSSION

Pancreatic cancer presents a grim prognosis primarily attributed to early lymph node infiltration or distant metastasis, resulting in a mere 10% 5-year survival rate (*Bengtsson, Andersson & Ansari, 2020*). Given this, there is a crucial need for non-invasive and highly accurate early diagnostic tools. CA19-9, a tumor-associated carbohydrate biomarker, holds

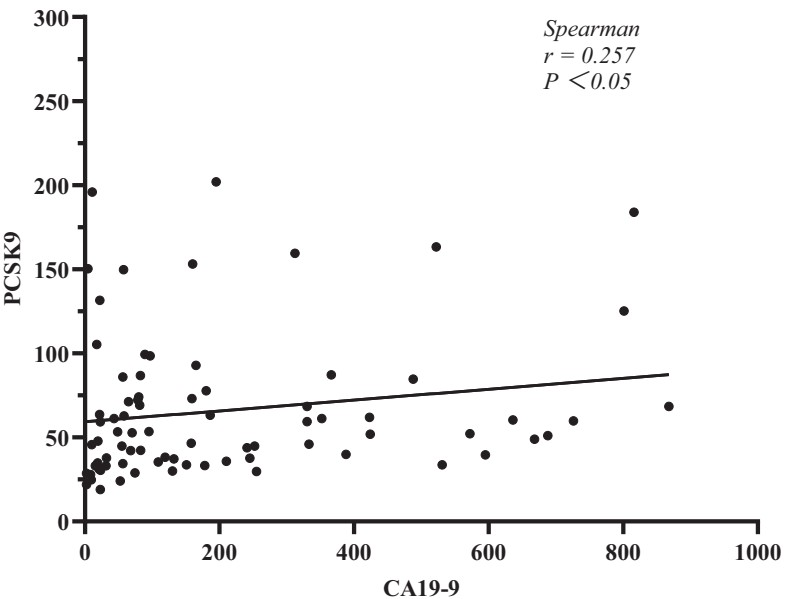

**Figure 3 Correlation analysis between PCSK9 and CA19-9 in PC patients.**

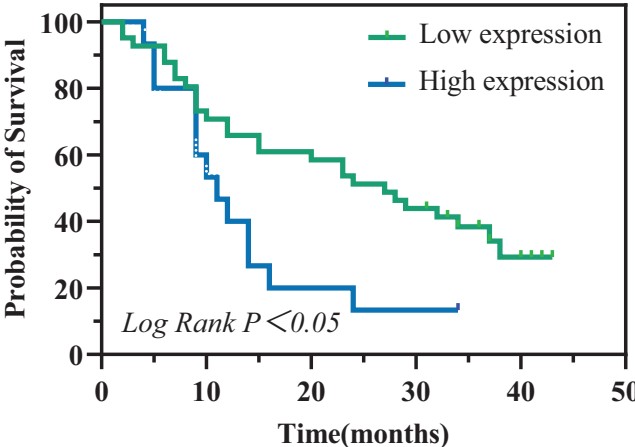

**Figure 4 Kaplan–Meier analysis of overall survival in patients with PC.**

pivotal diagnostic significance in pancreatic cancer. Its correlation with tumor mass, TNM stage, and recurrence underscores its clinical relevance (*Luo et al., 2021*). Despite its importance, CA19-9 exhibits limited effectiveness in early detection. Previous research suggests that its diagnostic accuracy could be enhanced through combination with additional markers (*Xu et al., 2022*), thereby addressing the imperative requirement for more reliable early detection strategies in pancreatic cancer.

Lipid metabolism is a significant metabolic alteration in cancer, involved in pancreatic cancer cellular proliferation, invasion, and metastasis (*Yin et al., 2022*). Research findings

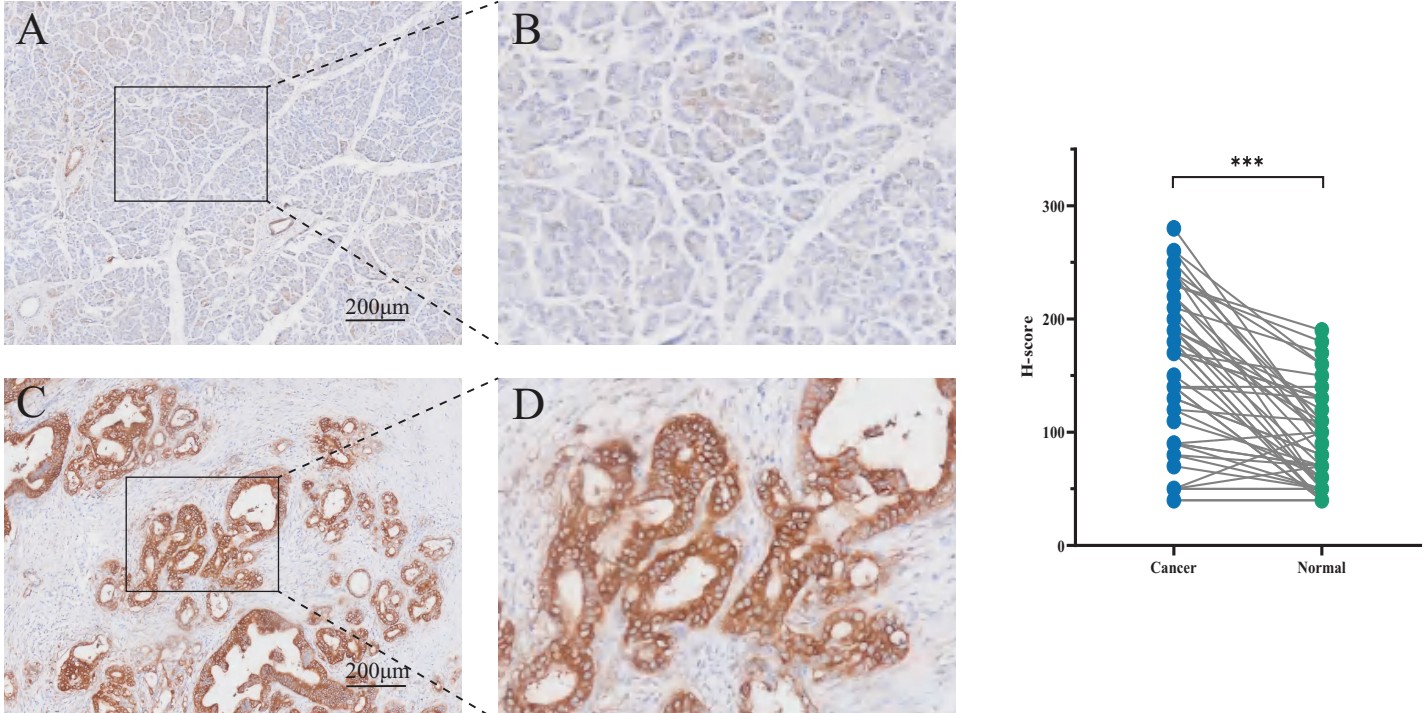

**Figure 5** **Immunohistochemistry (IHC) staining tissues of PCSK9 in pancreatic cancer.** (A, B) Expression of PCSK9 protein in human pancreatic normal tissue ($n = 48$). (C, D) Expression of PCSK9 protein in human pancreatic cancer tissue ($n = 48$). *$P < 0.05$, **$P < 0.01$ and ***$P < 0.001$. ns, none significance.

revealed that lipoprotein metabolic processes, particularly cholesterol uptake, are drastically activated in pancreatic tumors (*Guillaumond et al., 2015*). PCSK9 binds to the LDLR to regulate lipoprotein homeostasis and immune checkpoints in cancer (*Liu et al., 2020*; *Schulz, Schluter & Laufs, 2015*; *Yuan et al., 2021*). It has been shown that there was an increased amount of cholesterol and an overexpression of LDLR in pancreatic tumor cells (*Guillaumond et al., 2015*). Studies revealed that genetically increased levels of LDL-cholesterol were associated with the progression of PC (*Acier et al., 2021*; *Jung et al., 2021*). Clinically, LDLR expression was positively correlated with a higher risk of recurrence and a shorter survival time in PC patients (*Gonias et al., 2017*). These findings identify LDLR as a novel metabolic target to inhibit PC progression. We evaluated the potential application of sPCSK9 as a prognostic and diagnostic biomarker for PC in this study. Our study showed that sPCSK9 expression is low in patients with pancreatic disease, particularly in patients with early-stage PC. This is the opposite of other cancers that have been reported before. In pancreatic cancer, low sPCSK9 levels may be associated with high tissue LDLR expression. However, sPCSK9 detection was less effective in PC diagnosis than CA19-9. Combining sPCSK9 with CA19-9 could potentially enhance the diagnostic capability of CA19-9 for PC patients, which suggests that this combination might be a potential biomarker for PC diagnosis. Moreover, the higher sPCSK9 group was shown to

have a significantly shorter overall survival rate time. Interestingly, by IHC we found that PSCK9 is highly expressed in tumor tissues compared with normal pancreatic tissues. Considering that PCSK9 is a secreted protein, we speculate that this may be associated with an overexpression of LDLR in pancreatic tumor cells. Studies have shown that PCSK9 performs as a ligand that moves into the cell by binding to the LDLR to form a receptor-ligand complex (*Seidah & Prat, 2022*), so it seems plausible that PCSK9 is highly expressed in tumor tissue and low in the serum of PC patients. However, the detailed mechanisms need to be further excavated.

At present, surgery is the primary therapy for pancreatic cancer. However, the feasibility of surgery is limited to 15% to 20% of patients due to early metastasis and rapid disease progression (*Tempero et al., 2019*). Radiotherapy and chemotherapy are crucial components of pancreatic cancer treatment, but pancreatic cancer is insensitive to chemotherapy and radiotherapy (*Cao et al., 2021*). Furthermore, pancreatic cancer is recognized for its high resistance to immune responses (*Leinwand & Miller, 2020*), with minimal efficacy observed for single-agent immunotherapy. Several clinical trials of PCSK9 inhibitors in cancer have been carried out (*Oza & Kashfi, 2023*). According to the latest studies, inhibiting PCSK9 is a promising way to enhance immune checkpoint therapy for cancer (*Gao et al., 2023*; *Wang et al., 2022*; *Yang et al., 2023*). Regardless, it is an issue for future research to investigate the therapeutic potential of PCSK9 inhibitors in pancreatic cancer.

However, our study still has several limitations, such as study design, patient sample size, potential biases in patient selection, and follow-up time. Secondly, patients with other digestive carcinomas, such as cholangiocarcinoma or hepatocellular carcinoma, should be included to eliminate the possibility of false-positive results. Moreover, it was determined that *in vitro* and *in vivo* studies were required to confirm the biological activity of PCSK9 further and explore the relationship between PCSK9 and immunotherapy in pancreatic cancer.

## CONCLUSIONS

This is the first study to reveal the usefulness of the levels of sPCSK9 as biomarkers for PC. sPCSK9 was considerably lower in PC patients than in HC and was significantly higher than in BDG. The diagnostic performance of CA19-9 for PC patients could be improved by combining sPCSK9 with CA19-9. In addition, a higher level of serum PCSK9 in PC patients was also highly correlated with a worse prognosis, indicating the possibility of the utilization of the sPCSK9 level as a prognostic factor.

## ACKNOWLEDGEMENTS

The authors gratefully acknowledge Jun-Hong Han (State Key Laboratory of Biotherapy and Cancer Center, Frontiers Science Center for Disease-Related Molecular Network, West China Hospital, Sichuan University) for helpful discussions.

### Funding

This work was supported by the 1.3.5 Project for Disciplines of Excellence, West China Hospital (ZYJC21042), and Sichuan University for Qing Zhu. The funders had no role in study design, data collection and analysis, decision to publish, or preparation of the manuscript.

### Grant Disclosures

The following grant information was disclosed by the authors:
West China Hospital: ZYJC21042.
Sichuan University.

### Competing Interests

The authors declare that they have no competing interests.

### Author Contributions

- Ying Xu conceived and designed the experiments, performed the experiments, analyzed the data, prepared figures and/or tables, authored or reviewed drafts of the article, resources, and approved the final draft.
- Yongfeng Xu conceived and designed the experiments, performed the experiments, prepared figures and/or tables, authored or reviewed drafts of the article, resources, and approved the final draft.
- Yang Yang performed the experiments, authored or reviewed drafts of the article, resources, and approved the final draft.
- Zhiwei Zhang performed the experiments, authored or reviewed drafts of the article, resources, and approved the final draft.
- Qunli Xiong performed the experiments, authored or reviewed drafts of the article, resources, and approved the final draft.
- Qing Zhu conceived and designed the experiments, performed the experiments, authored or reviewed drafts of the article, and approved the final draft.

### Human Ethics

The following information was supplied relating to ethical approvals (*i.e.*, approving body and any reference numbers):

The study was approved by the Ethics Committee of West China Hospital of Sichuan University (1189, 20220325).

### Data Availability

The raw measurements are available in the Supplementary File.

## Supplemental Information

Supplemental information for this article can be found online at http://dx.doi.org/10.7717/peerj.18018#supplemental-information.

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
