# Peer review of "Serum PCSK9 is a novel serological biomarker for the diagnosis and prognosis of pancreatic cancer"

_PeerJ, doi:10.7717/peerj.18018_

## Round 0.1 · original submission · Major Revisions

The reviewers have identified many issues that need to be faced. To improve your manuscript, please try to correctly address all of them.

Reviewer 1 ·

Basic reporting

See Additional comments

Experimental design

See Additional comments

Validity of the findings

See Additional comments

Additional comments

Dear Editor
The manuscript (#99803) titled " Serum PCSK9 is a novel serological biomarker for the diagnosis and prognosis of pancreatic cancer " submitted for consideration in PeerJ. After careful review of the manuscript, I regret to inform you that I do not recommend its publication in PeerJ at this time. There are many problems could not be solved and outlined below.
1. The results of the authors show that the sPCSK9 in pancreatic cancer patients is between healthy control group and benign disease group, and this serum concentration range may lead to more false positive or false negative results. In addition, PCSK9 plays a major role in regulating cholesterol metabolism, and its elevated level is associated with an increased risk of a variety of cardiovascular diseases. Therefore, patients with underlying cardiovascular disease or other liver diseases may affect the PCSK9 level significantly rather than tumor patients.
2. The diagnostic difficulty of pancreatic cancer is how to detect the cancer at an early stage, although the authors showed that patients with early pancreatic cancer had considerably lower sPCSK9 levels than those with advanced pancreatic cancer. However, it is not helpful for the early diagnosis of pancreatic cancer. Late stage patients does not require any
3. The experimental design is too simple to reach the publication level. More experiments should be considered, such as immunohistochemistry and detecting the changes before and after surgery to seek any minimal residue disease, which is a biomarker’s goal.
Serological biomarkers of pancreatic cancer should have the characteristics of high specificity, high sensitivity, correlation with tumor status, easy detection, clear reference values and can be used in combination. The expression of a serological biomarker should be significantly higher in pancreatic cancer patients than in non-pancreatic cancer patients in order to accurately distinguish pancreatic cancer from other diseases. In addition, the normal and abnormal values of markers should be clearly defined so that doctors can determine whether a patient is at risk for pancreatic cancer. For example, the normal value of CA19-9 is usually less than 37 U/mL, and when this value is exceeded, the possibility of pancreatic cancer should be further considered. The authors’ results do not support the use of sPCSK9 as a novel serological biomarker for the diagnosis and prognosis of pancreatic cancer.
I encourage the authors to carefully consider those concepts and do more work on improving the experimental design and screening biomarkers for future submission.

Reviewer 2 ·

Basic reporting

This manuscript reports potential utility of sPCSK9 as diagnostic and prognostic biomarker o pancreatic ductal adenocarcinoma (PDAC). sPCSK9 has not been previously evaluated as a diagnostic PDAC biomarker. Although the prognostic significance of sPCSK9 has been established in several cancers, it has not been demonstrated in PDAC. Therefore, the study is innovative. The results are adequately presented, and the manuscript is written in a clear manner. There are several important omissions, that have to be corrected.
1. For each comparison, statistical significance of differences between AUC curves has to be calculated to enable authors to come to meaningful conclusions.
2. The purpose of comparison of PDAC vs HC is not clearly explained. If it is for early detection, then only early/resectable PDAC cases should be included in analysis. For the presented comparison of all stages PDAC group vs HC, sPCSK9 is unlikely to improve classification based on the graph. Therefore, the calculations have to be redone to a. include relevant cases, and b. present statistical significance of difference between CA19.9 and CA19.9+sPCSK9 AUCs.
3. For differential diagnosis of PDAC vs BDG, sPCSK9 seems to improve classification, but this could be definitively determined only after calculating statistical significance.
4. Since the discovery groups are relatively small, validation in independent sets is recommended.

Experimental design

The results of initial discovery of potential value of sPCSK9 for diagnosis and prognosis of PDAC are intriguing, but validation in independent set is recommended. Some comparisons and analyses have to be redone as described in Basic reporting.

Validity of the findings

The reported findings are innovative and hold translational potential. However, validation in independent sets is necessary for the results to be fully credible.

---

## Round 0.2 · accepted · Accept

We were unable to get the prior reviewers to re-review, however I have checked whether the authors have satisfactorily addressed the issues raised by the reviewers and I can confirm that this article is Acceptable now